# INDUCING GAUSSIAN PROCESS NETWORKS

## ABSTRACT

Gaussian processes (GPs) are powerful but computationally expensive machine learning models, requiring an estimate of the kernel covariance matrix for every prediction. In large and complex domains, such as graphs, sets, or images, the choice of suitable kernel can also be non-trivial to determine, providing an additional obstacle to the learning task. Over the last decade, these challenges have resulted in significant advances being made in terms of scalability and expressivity, exemplified by, e.g., the use of inducing points and neural network kernel approximations. In this paper, we propose inducing Gaussian process networks (IGN), a simple framework for simultaneously learning the feature space as well as the inducing points. The inducing points, in particular, are learned directly in the feature space, enabling a seamless representation of complex structured domains while also facilitating scalable gradient-based learning methods. We consider both regression and (binary) classification tasks and report on experimental results for real-world data sets showing that IGNs provide significant advances over state-of-the-art methods. We also demonstrate how IGNs can be used to effectively model complex domains using neural network architectures.

## 1 INTRODUCTION

Gaussian processes are powerful and attractive machine learning models, in particular in situations where uncertainty estimation is critical for performance, such as for medical diagnosis (Dusenberry et al., 2020).Whereas the original Gaussian process formulation was limited in terms of scalability, there has been significant progress in scalable solutions with Quiñonero-Candela & Rasmussen (2005) providing an early unified framework based on inducing points as a representative proxy of the training data. The framework by Quiñonero-Candela & Rasmussen (2005) has also been extended to variational settings (Titsias, 2009; Wilson et al., 2016b; Bauer, 2016), further enabling a probabilistic basis for reasoning about the number of inducing points (Uhrenholt et al., 2021). In terms of computational scalability, methods for leveraging the available computational resources have recently been considered (Nguyen et al., 2019b; Wang et al., 2019), with Chen et al. (2020) also providing insights into the theoretical underpinnings for gradient descent-based solutions in correlated settings (as in the case for GPs).

Common for most of the inducing points-based approaches to scalability is that the inducing points live in the same space as the training points (see e.g. (Snelson & Ghahramani, 2006; Titsias, 2009; Hensman et al., 2013; Damianou & Lawrence, 2013)). However, learning inducing points in the input space can be challenging for complex domains (e.g. over graphs), domains with high dimensionality (e.g. images), or domains with varying cardinality (e.g. text or point clouds) (Lee et al., 2019; Aitchison et al., 2021). More recently, methods for reasoning about the inducing points directly in embedding space have also been considered, but these methods often constrain the positions of the inducing points (Wilson et al., 2016a) or the structure of the embedding space (Lázaro-Gredilla & Figueiras-Vidal, 2009; Bradshaw et al., 2017), or they rely on complex inference/learning procedures (Aitchison et al., 2021)

In this paper, we propose inducing Gaussian process networks (IGN) as a simple and scalable framework for jointly learning the inducing points and the (deep) kernel (Wilson et al., 2016a). Key to the framework is that the inducing points live in an unconstrained feature space rather than in the input space. By defining the inducing points in the feature space together with an amortized pseudo-label function, we are able to represent the data distribution with a simple base kernel (such as

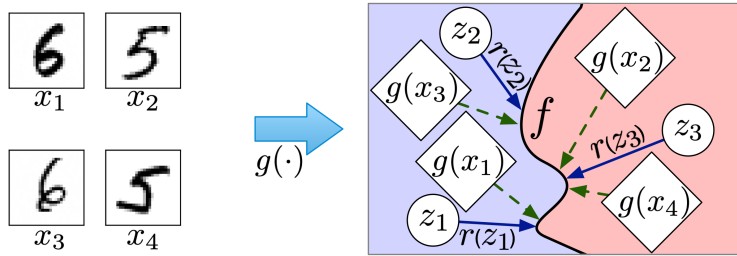

Figure 1: To the left four MNIST digits are embedded in the feature space by the neural network $\boldsymbol{g}$. To the right, the feature space where both features and inducing points exist. The observations associated with the inducing points $\boldsymbol{z}_1$, $\boldsymbol{z}_2$, and $\boldsymbol{z}_3$ are given by the pseudo-label function $r$, while the predictions associated with $\boldsymbol{g}(\boldsymbol{x}_1)$, $\boldsymbol{g}(\boldsymbol{x}_2)$, $\boldsymbol{g}(\boldsymbol{x}_3)$, and $\boldsymbol{g}(\boldsymbol{x}_4)$ are estimated using the GP posterior.

the RBF and dot-product kernel), relying on the expressiveness of the learned features for capturing complex interactions.

For learning IGNs, we rely on a maximum likelihood-based learning objective that is optimized using mini-batch gradient descent (Chen et al., 2020). This setup allows the method to scale to large data sets as demonstrated in the experimental results. Furthermore, by only having the inducing points defined in feature space, we can seamlessly employ standard gradient-based techniques for learning the inducing points (even when the inputs space is defined over complex discrete/hybrid objects) without the practical difficulties sometimes encountered when learning deep neural network structures.

We evaluate the performance of the proposed framework on several well-known data sets and show significant improvements compared to state of the art methods. We provide a qualitative analysis of the framework using a two-class version of the MNIST dataset. This is complemented by a more detailed quantitative analysis using the full MNIST and CIFAR10 data sets. Lastly, to demonstrate the versatility of the framework, we also provide sentiment analysis results for both a text-based and a graph-based dataset derived from the IMDB movie review dataset.

## 2 THE INDUCING GAUSSIAN PROCESS NETWORKS (IGN) FRAMEWORK

We start by considering regression problems, defined over an input space $\mathcal{X}$ and a label space of observations $\mathbb{R}$, modeled by a Gaussian process:

$$f \sim \mathcal{GP}(0, k(\cdot, \cdot)), \quad y = f(x) + \epsilon, \quad x \in \mathcal{X}, y \in \mathbb{R}, \tag{1}$$

where $k(\cdot, \cdot) : \mathcal{X} \times \mathcal{X} \to \mathbb{R}$ denotes the kernel describing the prior covariance, and $\epsilon \sim \mathcal{N}(0, \sigma_\epsilon^2)$ is the noise associated with the observations. We assume access to a set of data points $\mathcal{D} = \{(\boldsymbol{x}_i, y_i)\}_{i=1}^n$ generated from the model in Equation 1, and we seek to learn the parameters that define $k$ and $\sigma_\epsilon$ in order to predict outputs for new points $\boldsymbol{x}_*$ in $\mathcal{X}$. In what follows we shall use $X$ and $\boldsymbol{y}$ to denote $(\boldsymbol{x}_1, \ldots, \boldsymbol{x}_n)^{\mathrm{T}}$ and $(y_1, \ldots y_n)^{\mathrm{T}}$, respectively.

Firstly, we propose to embed the input points using a neural network $\boldsymbol{g}_{\boldsymbol{\theta}_g} : \mathcal{X} \to \mathbb{R}^d$ parameterized by $\boldsymbol{\theta}_g$. Secondly, for modeling the kernel function $k$, we introduce a set of $m$ inducing points $Z = (\boldsymbol{z}_1, \ldots, \boldsymbol{z}_m)^{\mathrm{T}}$, $\boldsymbol{z}_i \in \mathbb{R}^d$, together with a (linear) pseudo-label function $r_{\boldsymbol{\theta}_r} : \mathbb{R}^d \to \mathbb{R}$ parameterized by $\boldsymbol{\theta}_r$. We will use $\boldsymbol{r} = (r_{\boldsymbol{\theta}_r}(\boldsymbol{z}_1), \ldots, r_{\boldsymbol{\theta}_r}(\boldsymbol{z}_m))^{\mathrm{T}}$ to denote the evaluation of $r$ on $Z$, where $\boldsymbol{r}$ will play a rôle similar to that of inducing variables (Quiñonero-Candela & Rasmussen, 2005). In the remainder of this paper, we will sometimes drop the parameter subscripts $\boldsymbol{\theta}_g$ and $\boldsymbol{\theta}_r$ from $\boldsymbol{g}$ and $\boldsymbol{r}$ for ease of representation. An illustration of the model and the relationship between the training data and the inducing points can be see in Figure 1.

We finally define $k : \mathbb{R}^d \times \mathbb{R}^d \to \mathbb{R}$ as the kernel between pairs of vectors in $\mathbb{R}^d$. In particular, we denote with

$$(K_{XX})_{ij} = k(\boldsymbol{g}(\boldsymbol{x}_i), \boldsymbol{g}(\boldsymbol{x}_j)) \quad (K_{ZX})_{ij} = k(\boldsymbol{z}_i, \boldsymbol{g}(\boldsymbol{x}_j))$$
$$(K_{XZ})_{ij} = k(\boldsymbol{g}(\boldsymbol{x}_i), \boldsymbol{z}_j) \quad (K_{ZZ})_{ij} = k(\boldsymbol{z}_i, \boldsymbol{z}_j),$$

the four matrices corresponding to the kernel evaluations for all pairs of observed inputs and inducing points. The joint distribution over the function values at the observed inputs and the inducing points is now given as

$$\begin{bmatrix} \boldsymbol{y} \\ \boldsymbol{r} \end{bmatrix} \sim \mathcal{N}\left(\boldsymbol{0}, \begin{bmatrix} K_{XX} + \sigma_\epsilon^2 I & K_{XZ} \\ K_{ZX} & K_{ZZ} \end{bmatrix}\right), \tag{2}$$

where we slightly overload notation by using $\boldsymbol{r}$ to denote the function values associated with the inducing points, even though they do not correspond to real observations. For a given data set $\mathcal{D}$, our goal is to jointly learn $\boldsymbol{\theta} = \{Z, \boldsymbol{\theta}_g, \boldsymbol{\theta}_r, \sigma_\epsilon\}$ by considering the marginal likelihood:

$$p(\boldsymbol{y}|X, \boldsymbol{\theta}) = \mathcal{N}(\hat{\boldsymbol{y}}, K_{X|Z}), \tag{3}$$

where $\hat{\boldsymbol{y}}$ is the predictive mean:

$$\hat{\boldsymbol{y}} = K_{XZ}K_{ZZ}^{-1}\boldsymbol{r}_{\boldsymbol{\theta}_r} \tag{4}$$

and $K_{X|Z}$ is the posterior kernel given the inducing points, i.e.

$$K_{X|Z} = (K_{XX} + \sigma_\epsilon^2 I) - K_{XZ}K_{ZZ}^{-1}K_{ZX}. \tag{5}$$

Note that the kernel $K_{X|Z}$ is implicitly parameterized by $\boldsymbol{\theta}_g$ through the embedding function $\boldsymbol{g}_{\boldsymbol{\theta}_g}$. Furthermore, by jointly optimizing $\boldsymbol{\theta}_r$ together with the other model parameters we establish the connection between $\boldsymbol{f}$ and $\boldsymbol{r}$.

As in (Chen et al., 2020), we define our objective function in terms of the log-likelihood of the posterior of Equation 3:

$$\ell(\boldsymbol{\theta}; \mathcal{D}) = -\frac{1}{2}(\boldsymbol{y} - \hat{\boldsymbol{y}})^T K_{X|Z}^{-1}(\boldsymbol{y} - \hat{\boldsymbol{y}}) - \frac{1}{2}\log|K_{X|Z}| - \frac{n}{2}\log(2\pi), \tag{6}$$

which we can minimize wrt. the parameters $\boldsymbol{\theta}$ using mini-natch gradient descent with appropriate scaling factors for the gradients (Chen et al., 2020). The gradient of $\ell(\boldsymbol{\theta}; \mathcal{D})$ wrt. $\boldsymbol{\theta}$ can be expressed as:

$$\begin{aligned} \nabla_{\boldsymbol{\theta}}\, \ell(\boldsymbol{\theta}; \mathcal{D}) = &\, K_{X|Z}^{-1}(\boldsymbol{y} - \hat{\boldsymbol{y}})\nabla_{\boldsymbol{\theta}}\hat{\boldsymbol{y}} - \frac{1}{2}\, tr(K_{X|Z}^{-1}\nabla_{\boldsymbol{\theta}}K_{X|Z}) \\ &+ \frac{1}{2}K_{X|Z}^{-T}(\boldsymbol{y} - \hat{\boldsymbol{y}})(\boldsymbol{y} - \hat{\boldsymbol{y}})^T K_{X|Z}^{-T}\nabla_{\boldsymbol{\theta}}K_{X|Z}, \end{aligned} \tag{7}$$

where $tr(A)$ represents the trace of a matrix $A$. Note that in Equation 7 both $\hat{\boldsymbol{y}}$ and $K_{X|Z}$ depend on $\boldsymbol{\theta}$ and that the joint optimization of the model parameters, in particular $\boldsymbol{\theta}_r$, establishes the connection between the Gaussian process and the pseudo-label function.

Based on the chain rule of differentiation, we see that the gradient of Equation 7 with respect to the inducing points $Z$ and the parameters $\boldsymbol{\theta}_r$ of the pseudo-label function do not depend on $\nabla\boldsymbol{g}_{\boldsymbol{\theta}_g}$, that is the gradients of $\boldsymbol{g}_{\boldsymbol{\theta}_g}$ have no impact on the updates of $Z$ and $\boldsymbol{\theta}_r$.

**Proposition 1.** *The gradients $\nabla_Z\ell(\boldsymbol{\theta}; \mathcal{D})$ and $\nabla_{\boldsymbol{\theta}_r}\ell(\boldsymbol{\theta}; \mathcal{D})$ do not depend on $\nabla\boldsymbol{g}_{\boldsymbol{\theta}_g}$.*

This is a key advantage of IGNs as learning the inducing points is therefore not influenced by the gradient of a potentially complex embedding function $\boldsymbol{g}$ and the entailed optimization difficulties. Additionally, as the inducing points (only) exists in $\mathbb{R}^d$, the underlying learning framework for the inducing points is indifferent to the structure of the input space $\mathcal{X}$ and whether it is discrete or continuous or defined over, e.g., graphs or images.

Once the IGN parameters $\boldsymbol{\theta} = \{Z, \boldsymbol{\theta}_g, \boldsymbol{\theta}_r, \sigma_\epsilon\}$ have been learned, we can find the predictive distribution for input points $X_*$ and inducing points $Z$, by first considering the joint distribution over the associated function values

$$\begin{bmatrix} \boldsymbol{r} \\ \boldsymbol{f}_* \end{bmatrix} \sim \mathcal{N}\left(\boldsymbol{0}, \begin{bmatrix} K_{ZZ} & K_{ZX_*} \\ K_{X_*Z} & K_{X_*X_*} \end{bmatrix}\right),$$

which in turn gives the predictive distribution

$$p(\boldsymbol{f}_*|X_*, \boldsymbol{\theta}) = \mathcal{N}\left(\hat{\boldsymbol{f}}_*, K_{X_*|Z}\right), \tag{8}$$

where

$$\hat{\boldsymbol{f}}_* = K_{X_*Z}K_{ZZ}^{-1}\boldsymbol{r} \qquad K_{X_*|Z} = K_{X_*X_*} - K_{X_*Z}K_{ZZ}^{-1}K_{ZX_*}. \tag{9}$$

Finally, IGNs can straightforwardly be extended to solve classification tasks. For ease of exposition, we only consider binary classification problems, but the general method can straightforwardly be extended to a multi-valued setting.[1] That is, we assume that the label space is given by $\mathcal{Y} = \{0, 1\}$ and that we have access to a data set $\mathcal{D} = \{(\boldsymbol{x}_i, y_i)\}_{i=1}^{n}$, where the underlying data generating process is defined by a latent function $f(\boldsymbol{x})$ with a GP prior

$$f \sim \mathcal{GP}(0, k(\cdot, \cdot))$$

and $y|\boldsymbol{x} \sim \Phi(f(\boldsymbol{x}))$, where $\Phi(\cdot)$ is the cumulative Gaussian function; see (Rasmussen & Williams, 2006). In order to perform maximum likelihood estimation of the model parameters, we approximate the likelihood using a Laplace approximation. For the sake of compactness, we defer the full derivation to Appendix B.

## 2.1 COMPUTATIONAL COMPLEXITY

The main aspect related to the computational complexity of the IGN framework concerns the kernel computation $K_{Z|X}$. If $n_z$ is the number of inducing points and $b$ is the mini-batch size (recall that for training we use mini-batch gradient descent), the complexity of computing the kernel is cubic wrt $n_z$, i.e. $\mathcal{O}(n_z^3)$. However, at inference time, we can omit the term connected to $K_{ZZ}^{-1}$, since it only needs to be efficiently computed once (Sharma et al., 2013). If we at this phase further assume that $n_z > b$, the complexity reduces to the computation of $K_{ZX}$ and $K_{XX}$, i.e. $\mathcal{O}(n_z \cdot b)$.

## 3 RELATED WORKS

The combination of kernels and neural networks has previously been explored, most notably in the context of deep kernel learning (Wilson et al., 2016a). Using a a deep neural network architecture, (Wilson et al., 2016a) transform the input vectors into feature space based on which a base kernel is applied (here the RBF kernel and the spectral mixture base kernel (Wilson & Adams, 2013) are used). The kernel parameters and the neural network weights are jointly learned by maximizing the marginal likelihood of the Gaussian process, but relying on a pre-training of the underlying deep neural network architecture. This work has been subsequently extended into a variational setting (Wilson et al., 2016b), also providing support for multi-class classification but again relying on a somewhat cumbersome pre-training.

A key difference between our approach and (Wilson et al., 2016a;b) is in our choice of inducing points. For instance, in (Wilson et al., 2016a;b), the inducing points are placed on a regular multi-dimensional lattice based on which the deep kernel is evaluated (van Amersfoort et al., 2022). In IGNs the inducing points are (only) defined in feature space, where they are treated as unconstrained parameters and learned jointly together with the neural network weights and (any) kernel parameters. The pseudo-label function of IGNs plays a role similar to that of inducing variables (Wilson et al., 2016b), but where each inducing variable is defined by its own set of variational parameters, the pseudo-label function offers an amortized approach by having the parameters of the pseudo-label function being shared among all inducing points. Furthermore, as shown in Section 4, our learning scheme is end-to-end and does not require any pre-training of networks as in (Wilson et al., 2016a). Finally, IGNs rely on mini-batch maximum likelihood-based gradient descent (Chen et al., 2020), thus avoiding the GP-KISS kernel approximation of (Wilson et al., 2016a) and the grid layout of the inducing points together with the variational-based sampling setup of (Wilson et al., 2016b).

Several other related works exploit inducing points. For instance, (Titsias, 2009; Hensman et al., 2013; Damianou & Lawrence, 2013) propose to maximize a lower bound of the exact marginal likelihood to learn the inducing points in the input space, in contrast to our IGN framework where the inducing points are defined in a (transformed) feature space. Using inducing point in a transformed is also explored in, e.g., (Lázaro-Gredilla & Figueiras-Vidal, 2009), but here the transformed space is fixed, e.g., in the form of the frequency domain of the input space. In our IGN framework, we instead use a transformation that is jointly learned together with a (deep) kernel; moreover, our inducing points can (in principle) also live in a space different from the embedding space. Closely related to IGNs is (Snelson & Ghahramani, 2006), where inducing points are jointly learned with kernel

---

[1]For the experiment results, (see Section 4.3) we have used a one-vs-all approach to solve the multi-class classification tasks.

Table 1: Comparison of root mean square error (RMSE) and negative log-likelihood (NLL) of different GPs on the benchmark datasets. The best results are in bold (lower is better).

| Dataset | IGN (Ours) | | sGGP | | DKL | | DIWP | |
|---|---|---|---|---|---|---|---|---|
| | RMSE | NLL | RMSE | NLL | RMSE | NLL | RMSE | NLL |
| Levy | $\mathbf{0.17}_{\pm\mathbf{0.01}}$ | $\mathbf{0.98}_{\pm\mathbf{0.06}}$ | $0.27_{\pm0.00}$ | $4.59_{\pm0.93}$ | $0.26_{\pm0.03}$ | $1.26_{\pm0.06}$ | $0.60_{\pm0.03}$ | $1.26_{\pm0.02}$ |
| Griewank | $\mathbf{0.05}_{\pm\mathbf{0.00}}$ | $\mathbf{0.76}_{\pm\mathbf{0.01}}$ | $0.07_{\pm0.00}$ | $0.87_{\pm0.01}$ | $0.15_{\pm0.07}$ | $1.44_{\pm0.27}$ | $0.08_{\pm0.02}$ | $0.93_{\pm0.00}$ |
| Borehole | $\mathbf{0.00}_{\pm\mathbf{0.00}}$ | $\mathbf{0.41}_{\pm\mathbf{0.03}}$ | $0.17_{\pm0.00}$ | $1.84_{\pm0.10}$ | $0.12_{\pm0.03}$ | $1.14_{\pm0.29}$ | $0.04_{\pm0.02}$ | $0.93_{\pm0.00}$ |
| Protein | $\mathbf{0.64}_{\pm\mathbf{0.01}}$ | $1.19_{\pm0.04}$ | $0.66_{\pm0.01}$ | $\mathbf{0.91}_{\pm\mathbf{0.42}}$ | $0.99_{\pm0.32}$ | $1.13_{\pm0.05}$ | $0.88_{\pm0.01}$ | $1.39_{\pm0.02}$ |
| PM2.5 | $0.31_{\pm0.01}$ | $\mathbf{1.16}_{\pm\mathbf{0.03}}$ | $\mathbf{0.29}_{\pm\mathbf{0.00}}$ | $7.10_{\pm3.24}$ | $0.63_{\pm0.31}$ | $2.32_{\pm6.75}$ | $0.82_{\pm0.05}$ | $1.35_{\pm0.08}$ |
| Energy | $0.74_{\pm0.02}$ | $\mathbf{1.02}_{\pm\mathbf{0.01}}$ | $0.79_{\pm0.00}$ | $1.24_{\pm0.12}$ | $0.71_{\pm0.03}$ | $1.21_{\pm0.03}$ | $\mathbf{0.68}_{\pm\mathbf{0.01}}$ | $1.21_{\pm0.01}$ |
| Bike-hour | $\mathbf{0.01}_{\pm\mathbf{0.00}}$ | $\mathbf{0.28}_{\pm\mathbf{0.06}}$ | $0.22_{\pm0.00}$ | $0.51_{\pm0.07}$ | $0.13_{\pm0.03}$ | $0.96_{\pm0.08}$ | $0.60_{\pm0.06}$ | $1.21_{\pm0.03}$ |
| Query | $0.07_{\pm0.01}$ | $\mathbf{0.77}_{\pm\mathbf{0.02}}$ | $\mathbf{0.05}_{\pm\mathbf{0.00}}$ | $1.97_{\pm0.09}$ | $0.07_{\pm0.03}$ | $1.16_{\pm0.36}$ | $0.41_{\pm0.01}$ | $1.34_{\pm0.04}$ |

parameters using gradient descent. However, in contrast to IGNs, the continuous optimization of $Z$ proposed by Snelson & Ghahramani (2006) is considerably more simplified as the inducing points are learned in the input space. Similarly to the proposed framework, Aitchison et al. (2021) exploit inducing Gram matrix. The Gram matrix are used to sample inducing points in the feature space, and subsequent GP predictions are conditionally sampled on the inducing points in the features space. This process, in contrast to IGNs, relies on a complex doubly-stochastic variational inference process.

The papers cited above are representatives of GP methods that are methodologically related to the proposed IGN framework. Not all of the cited methods are, however, in line with state of the art in terms of, e.g., accuracy results, hence the experimental section below also includes descriptions and comparisons of other GP methods, which forms the basis for the empirical evaluation and analysis.

## 4 Experiments

In this section, we present and discuss experimental results showing the potential of the IGN framework on both regression and classification tasks. Both settings are aimed at showing the ability of IGNs to jointly learn the inducing points and the model parameters. In the experiments, we also compare IGNs against other state of the art methods that either learn the kernel parameters or the inducing points. The variety of the experiments aim to provide insight into the properties of the framework and show the potential of IGNs for diverse real-world datasets, outperforming state-of-the-art methods in the majority of the cases.

### 4.1 Regression tasks

Following the setup described in (Chen et al., 2020), we compared our approach to deep kernel learning (DKL) (Wilson et al., 2016a), deep kernel processes (DIWP) (Aitchison et al., 2021), and sgGP (Chen et al., 2020) on several simulated and real regression benchmark datasets. Briefly, DKL exploits inducing points (in the embedding space) and an approximation of the kernel to scale to large datasets, DIWP defines deep kernel processes where positive definite Gram matrices are progressively transformed by nonlinear kernel functions and by sampling from (inverse) Wishart distributions.

The sizes and feature dimensions for each dataset are reported in Appendix (see Table 3). The simulated datasets (Levy, Griewank and Borehole) are from the Virtual Library of Simulation Experiments[2], while the real datasets (Protein, PM2.5, Energy, Bike-hour, and Query) are from the UCI repository[3].

For each dataset, we repeated the experiments 10 times. In each experiment, the dataset was randomly split into a 60% training set and a 40% test set. Furthermore, the training set was normalized to have 0 mean and 1 standard deviation, and the test set was scaled accordingly. For all the datasets, we considered a simple neural network with 3 stacked dense layers with 128 units and ReLU activations.

---

[2] https://www.sfu.ca/~ssurjano/
[3] http://archive.ics.uci.edu/ml

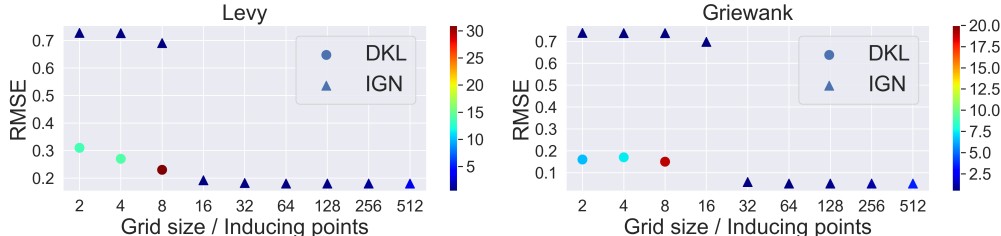

Figure 2: RMSE of DKL (circles) and IGN (triangles) for Levy and Griewank datasets in function of the grid size (for DKL) and number of inducing points (for IGN). The colors encode the time (in seconds) that each epoch takes to be completed. DKL RMSE improves with the grid size but the computational time drastically increases. On the other hand, IGN requires more inducing points but the results are overall better and the computational time remains stable. All the models were run on CPU for having a fair comparison.

On top of those we added a final feature layer with 64 units. We used 512 inducing points[4] having the same dimension as the feature layer (i.e., 64 dimensions somewhat arbitrarily chosen), an RBF kernel with $\gamma = 1.0$ for all dimensions, and a linear pseudo-label function. The model weights are updated with the Adam optimizer for 500 epochs on mini-batches of size 128.

Table 1 summarizes the results, where for each dataset we report the average and standard deviation of the root mean squared error and negative log-likelihood on the test set over the 10 experimental repetitions. For Levy, Borehole, and Bike-hour datasets, our approach outperforms by a large margin the other methods. While for PM2.5, Energy, and Query datasets our approach performed slightly worse than the others, we still obtained comparable results. Overall, our method remained stable as the standard deviations over 10 experiments are small and comparable to sgGP. Additional results using other baseline methods can be found in Appendix C.

Finally, for illustrating the effect of imposing a grid structure over the inducing points wrt. computation time and accuracy, we have made a comparison of IGN and DKL (Wilson et al., 2016a). Recall that DKL requires a $n$-dimensional grid whose cost grows exponentially with the amount of data. In this experiments, we use the exact same architecture in terms of layers and kernel as described in Section 4.1 for both IGN and DKL. Figure 2 shows the root mean square error for Levy and Griewank datasets. Our method, although requiring more inducing points, outperforms DKL in terms of computational time and root mean square errors (evaluated on the test sets). It was not possible to conduct experiments for DKL with more than 8 inducing points as we hit the memory limit.

## 4.2 Toy-MNIST

Toy-MNIST is the same as MNIST, but limited to digits belonging to class 5 or 6. The aim of this experiments is to show the ability of IGNs to classify relatively ambiguous digits, and provide insights into the inducing points and the uncertainty associated with the predictions. In this case, the training and test sets consist of 11,339 and 1,850 images, respectively. The image labels associated with classes 5 and 6 are replaced with 1 and -1, respectively, and we assume the exact same model specification as in Section 4.1, but now only using 64 inducing points.

By repeating the experiment 10 times we obtain a test set accuracy of $99.25 \pm 0.09\%$. For each of the 64 inducing points, we retrieve the closest image in the training set based on the chosen kernel in feature space, after which we refine each of the images (relative to the distance to the closest inducing point) using gradient descent with respect to the input image. The 64 inducing points (in the feature space) and digit images corresponding to the inducing points are shown in Figure 3. For this experiment, the intuitive interpretation of the images associated with the inducing point also provides a degree of explainabilty, even though the inducing points are learned in feature space. As with standard Gaussian processes, IGNs retains the feature of uncertainty estimation. This is illustrated in Figure 4, which depicts a scatter plot of the images in the test set, given in terms of the expected

---

[4]We randomly initialized the inducing points with same initializer as the final feature layer, i.e. Xavier uniform initializer.

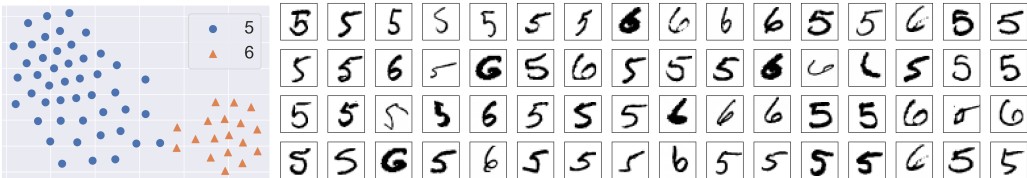

Figure 3: To the left the 64 inducing points projected into two-dimensional space. To the right the 64 pseudo images corresponding to the 64 inducing points for the Toy-MNIST experiment.

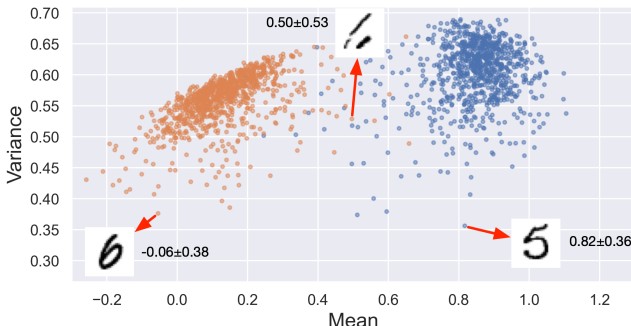

Figure 4: Mean (x-axis) against variance (y-axis) plot for the Toy-MNIST dataset. Orange and blue dots correspond to images labeled with 6 and 5, respectively. Images associated with low variances and high mean (in absolute value) correspond to clear digits. On the other hand, images with high variance and mean close to the decision boundary, i.e. 0.5, correspond to ambiguous digits.

value of the predictions (x-axis) and the variance of the predictions (y-axis). The two distinct orange and blue clusters contain the digits corresponding to classes 6 and 5. The figure also illustrates some of the extreme predictions: the two digits with means far from 0.5 and low variance correspond to clearly distinguishable digits, while the digit with a mean close to 0.5 and relatively high variance corresponds to a more ambiguous image.

## 4.3 MNIST AND CIFAR10

In this section we evaluate IGN on computer vision classification tasks. Specifically, here we focus on MNIST and CIFAR10. We compare our approach against deep kernel processes (DIWP) (Aitchison et al., 2021) as well as Deep Gaussian Processes (DGP) (Damianou & Lawrence, 2013), NNGP (Blundell et al., 2015), and stochastic variational deep kernel learning (SVI-DKL) (Wilson et al., 2016b). Briefly, DGP uses graphical models to nest layers of Gaussian processes, and NNGP adopts a back-propagation-compatible algorithm for learning a probability distribution on the weights of a neural network to estimate the uncertainty of the model.

For the sake of simplicity , we decompose with a one-vs-all approach each multiclass task into 10 binary tasks. We again used the same model as in Section 4.1 (this model is comparable to the other competitor models in terms of number of parameters), but with 32 and 16 inducing points for MNIST and CIFAR10, respectively (the same number of inducing points was used by all methods). For each of the datasets, we repeated the experiments 10 times and report the average accuracy and negative log-likelihood in Table 2. We observe that our method outperforms all the others in terms of accuracy, while keeping the standard deviation comparably small. The negative log-likelihood is better for CIFAR-10, but worse for MNIST, suggesting that training a full multi-classification model could be beneficial, see (Murphy, 2022).

Table 2: Test set accuracy (ACC) and negative log-likelihood (NLL) comparison of MNIST and CIFAR10 for IGN approach against DGP, NNGP, DIWP, and SVI-DKL.

| DATASET | IGN (OURS) | | DGP | | NNGP | | DIWP | | SVI-DKL | |
|---|---|---|---|---|---|---|---|---|---|---|
| | ACC | NLL | ACC | NLL | ACC | NLL | ACC | NLL | ACC | NLL |
| MNIST | $\mathbf{98.0}_{\pm \mathbf{0.0}}$ | $0.6_{\pm 0.0}$ | $96.5_{\pm 0.1}$ | $0.1_{\pm 0.0}$ | $96.5_{\pm 0.0}$ | $0.1_{\pm 0.0}$ | $97.7_{\pm 0.0}$ | $\mathbf{0.1}_{\pm \mathbf{0.0}}$ | $97.4_{\pm 0.1}$ | $1.3_{\pm 0.0}$ |
| CIFAR10 | $\mathbf{51.0}_{\pm \mathbf{0.2}}$ | $\mathbf{1.1}_{\pm \mathbf{0.0}}$ | $46.8_{\pm 0.1}$ | $1.5_{\pm 0.0}$ | $47.4_{\pm 0.1}$ | $1.5_{\pm 0.0}$ | $50.5_{\pm 0.1}$ | $1.4_{\pm 0.0}$ | $48.1_{\pm 0.5}$ | $1.6_{\pm 0.0}$ |

## 4.4 STRUCTURED DATASETS

In this section, we investigate the potential of our model applied to datasets with more complex structure. Specifically, we consider two classification tasks on text and graphs, respectively. The datasets for both tasks are derived from IMDB:

- IMDB-TEXT (Maas et al., 2011) consists of 25,000 training reviews and 25,000 test reviews. Positive and negative labels are balanced within the training and test sets. We preprocess the data by keeping the top 20,000 words and limiting the length of words in a review to 200.

- IMDB-GRAPH is a social network dataset contained in the collection described in (Yanardag & Vishwanathan, 2015). It consists of constructed genre-specific collaboration networks where nodes represent actresses/actors who are connected by an edge if they have appeared together in a movie of a given genre. Collaboration networks are generated for the genres Action and Romance for this dataset. The data then consists of the ego-graphs for all actresses/actors in all genre networks, and the task is to identify the genre from which an ego-graph has been extracted. It contains 1,000 graphs with labels balanced among the two classes.

For both experiments, we aim to show that our approach can jointly handle complex data and complex models for extracting features, hence the aim is not to compare against state-of-the-art methods. In particular, the two IGN instantiations for the two datsets differ only in the embedding functions/feature extractors being used. For both datasets, we also provide baseline accuracies obtained by two classifiers sharing the architectures of the feature extractors in the IGNs.

For IMDB-TEXT, we used 128 inducing points and trained a transformer-based model (Vaswani et al., 2017) to learn features from the reviews. We repeated the experiments 10 times for the baseline and our approach obtaining accuracies of $84.35\% \pm 0.10$ and $84.43\% \pm 0.96$, respectively. Similarly to Section 4.2, we can also inspect the learned inducing points. For example, included below are the two training set examples (positive and negative) that are closets to two randomly chosen learned inducing points.[5]

- *I was literally preparing to hate this movie so believe me when I say this film is worth seeing ... my score 7 out of 10.*

- *I can't believe that so many are comparing this movie to Argento's ... If you're looking for a good horror movie look elsewhere.*

Similar to Section 4.2 we also here see that by associating the learned inducing points with training instances provides an additional element of explainability to the inducing points.

For the IMDB-GRAPH, we follow the setup described by Tibo et al. (2020) and represent each graph as a set of neighborhoods. We performed a 10 times 10 fold cross-validation for both the baseline and IGN. We ran 100 epochs of the Adam optimizer with learning rate 0.001 on mini-batches of size 32. We obtained accuracies equal to $72.49\% \pm 0.60$ and $73.40\% \pm 0.63$, for the baseline and IGN, respectively. Without any particular effort, in terms of fine tuning model structure and parameters, our method outperforms most of the graph neural networks reported in (Nguyen et al., 2019a).

---

[5]More correspondences between training set reviews and inducing points are reported in the supplementary material.

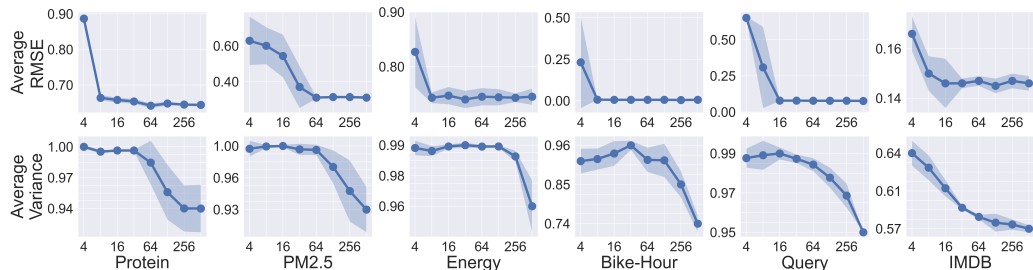

Figure 5: Average RMSE (first row) and variance (second row) estimated on 10 runs for PROTEIN, PM2.5, ENERGY, BIKE-HOUR, QUERY, and IMBD-TEXT. For IMDB we report the classification error. The variance constantly decreases with the number of inducing points for all the datasets.

## 5 ABLATION ON NUMBER OF INDUCING POINTS

Here we study the impact of the number of inducing points for the regression and classification tasks defined in the previous sections. The variance of Gaussian processes estimations decrease with the number of training points (Rasmussen & Williams, 2006). In our case, the training points are replaced by inducing points but we retain similar result. For the sake of completeness, we slightly reformulate the proposition and provide a proof in the supplementary material.

**Proposition 2.** *The variance of a test point $(\boldsymbol{x}_*, f_*)$ given a set of inducing points $Z$ can never increase by the inclusion of an additional inducing point $\boldsymbol{z} \notin Z$.*

When training IGNs with different number of inducing points, there are no guarantees that the same subset of inducing points will be learned during training. However, we still observe a consistent decrease in variance when the number of inducing points increases. For the regression tasks, we report here the results for all the real datasets described in Section 4.1. For the classification task we focus on IMDB-TEXT described in Section 4.4. In all the cases, we repeated the experiments 10 times varying the number of inducing points (4,8,16,32,64,128,256,512). Figure 5 depicts the root mean square error (on the top), and the average variance with error bars (on the bottom) for the different datasets. From the figure we see that the average variance consistently decreases with the number of inducing points for all datasets. For some of the datasets with only a few inducing points, the error bars are lowe, but the root mean square error is always higher. Finally, it is worth noticing that even when the root mean square error stabilizes, the variance keeps decreasing with the number of inducing points.

## 6 CONCLUSIONS AND FUTURE WORK

While Gaussian processes are powerful machine learning methods that can estimate uncertainty, they remain intractable on large datasets. Several works have addressed this problem using inducing points but, to the best of our knowledge, these methods remain limited to comparatively simple datasets. In this paper, we introduce a framework for learning Gaussian processes for large and complex datasets by combining inducing points in feature space with neural network kernel approximations. We empirically showed the ability of our method on standard machine learning benchmarks as well as on structured (graph and text) datasets, with the proposed method outperforming other state-of-the-art approaches.

The flexibility of our IGN framework enables the combination of Gaussian process training with complex deep learning architectures. We believe that our method provides a basis for further research in calibration of uncertainty estimates in deep neural network models. Furthermore, as part of future work, we also plan to investigate how to position IGN within a fully variational setting (Titsias, 2009), exploring the hierarchical structure of the model, possibly in the context of deep kernel processes (Aitchison et al., 2021).

## ETHICS STATEMENT

The work reported in this paper is not related to human subjects, practices to data set releases, discrimination/bias/fairness concerns, and research integrity issues. The work is also not related to any particular application domain. The paper proposes a simple but general framework for learning inducing points in sparse Gaussian processes, also contributing with insights/explainability into the model (through the learned inducing points) and potentially improved accuracy and uncertainty estimates. Depending on the application domain, the significance and interpretation of these insights and estimates may require careful scrutiny by domain experts, in particular for safety critical applications.

## REPRODUCIBILITY STATEMENT

An open-source implementation of our IGN framework has been uploaded as supplementary material together with the paper. Upon acceptance of the paper, the code will be placed in a public GitHub repository. The implementation also contains detailed information about the data sets, parameters, and random seeds used in the experiments.

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

## A    ABOUT THE VARIANCE OF INDUCING POINTS

**Proposition 2.**  *The variance of a test point $(\boldsymbol{x}_*, f_*)$ given a set of inducing points $Z$ can never increase by the inclusion of an additional inducing point $\boldsymbol{z} \notin Z$.*

*Proof.*  Let $S$ and $R$ be two sets containing inducing points. Let $K_{f_*,\boldsymbol{s}|\boldsymbol{r}}$ be the variance associated with $f_*$ conditioned only on the inducing point in $R$. Let $K_{f_*|\boldsymbol{s},\boldsymbol{r}}$ be the variance associated with $f_*$ conditioned only on the inducing point in $R \cup S$. If $K'_{X_* X_*}$ is the top-left scalar in $K_{f_*,\boldsymbol{s}|\boldsymbol{r}}$ representing the variance of $f_*$, then we prove that $K_{f_*|\boldsymbol{s},\boldsymbol{r}} \leq K'_{X_* X_*}$. Let $K$ be the covariance matrix of the Gaussian process having the following block structure:

$$K = \begin{bmatrix} K_{X_* X_*} & K_{X_* R} & K_{X_* S} \\ K_{RX_*} & K_{RR} & K_{RS} \\ K_{SX_*} & K_{SR} & K_{SS} \end{bmatrix},$$

where the subscripts represents the two sets (among the observation $\{\boldsymbol{x}_*\}$ and inducing points in $R$ and $S$) used to evaluate the kernel matrix. We compute now $K_{f_*,\boldsymbol{s}|\boldsymbol{r}}$ as

$$\begin{aligned} K_{f_*,\boldsymbol{s}|\boldsymbol{r}} &= \begin{bmatrix} K_{X_* X_*} & K_{X_* S} \\ K_{SX_*} & K_{SS} \end{bmatrix} - \begin{bmatrix} K_{X_* R} \\ K_{SR} \end{bmatrix} K_{RR}^{-1} \begin{bmatrix} K_{RX} K_{RS} \end{bmatrix} \\ &= \begin{bmatrix} K'_{X_* X_*} & K'_{X_* S} \\ K'_{SX_*} & K'_{SS} \end{bmatrix}. \end{aligned} \tag{10}$$

Note that $K'_{X_* X_*} \leq K_{X_* X_*}$. Finally,

$$K_{f_*|\boldsymbol{s},\boldsymbol{r}} = K'_{X_* X_*} - K'_{X_* S} K_{SS}'^{-1} K'_{SX_*} \leq K'_{X_* X_*}$$

which concludes the proof.  $\square$

## B    IGNs FOR CLASSIFICATION

Here we provide additional details about the full derivation (see Section 2.1 in the main paper) of IGNs for classification. For ease of exposition, we only consider binary classification problems, but the general method can straightforwardly be extended to a multi-valued setting. For the experiment results, (see Section 4.3 in the main paper) we therefore use a one-vs-all approach to solve multiclass classification tasks. We assume that the label space is given by $\mathcal{Y} = \{0, 1\}$ and that we have access to a data set $\mathcal{D} = \{(\boldsymbol{x}_i, y_i)\}_{i=1}^n$, where the underlying data generating process is defined by a latent function $f(\boldsymbol{x})$ with a GP prior

$$f \sim \mathcal{GP}(0, k(\cdot, \cdot))$$

and $y|\boldsymbol{x} \sim \Phi(f(\boldsymbol{x}))$, where $\Phi(\cdot)$ is the cumulative Gaussian function; see (Rasmussen & Williams, 2006).

For inference, we calculate the posterior distribution $p(\boldsymbol{y}|X_*, \boldsymbol{\theta})$ given by

$$p(\boldsymbol{y}|X_*, \boldsymbol{\theta}) = \int p(y|\boldsymbol{f}_*) p(\boldsymbol{f}_*|X_*, \boldsymbol{\theta}) d\boldsymbol{f}_* .$$

Assuming that $\boldsymbol{f}_* \sim \mathcal{N}(\mu, \sigma^2)$ for some $\mu$ and $\sigma$ we get (Rasmussen & Williams, 2006, Section 3.9):

$$p(y_*|\boldsymbol{x}_*, \boldsymbol{\theta}) = \Phi(\alpha) \text{ with } \alpha = \frac{\mu}{\sqrt{1 + \sigma^2}},$$

where $\mu$ and $\sigma^2$ can be found from the predictive distribution in Equation 8 (in the main paper).

For learning the IGN parameters $\boldsymbol{\theta}$ we perform maximum likelihood estimation, but using the Laplace approximation to derive an approximation of the marginal likelihood. Firstly, the likelihood is defined as

$$p(\boldsymbol{y}|X, \boldsymbol{\theta}) = \int p(\boldsymbol{y}|\boldsymbol{f}) p(\boldsymbol{f}|X, \boldsymbol{\theta}) d\boldsymbol{f}. \tag{11}$$

Denoting by

$$\hat{\boldsymbol{f}} = \arg\max_{\boldsymbol{f}} \log p(\boldsymbol{y}|\boldsymbol{f}) + \log p(\boldsymbol{f}|X, \boldsymbol{\theta})$$

and employing the Laplace approximation in Equation 11, our goal is to maximize the following log-likelihood approximation

$$\ell(\boldsymbol{\theta}; \mathcal{D}) \approx \log p(\boldsymbol{y}|\hat{\boldsymbol{f}}) + \log p(\hat{\boldsymbol{f}}|X, \boldsymbol{\theta}) - \frac{1}{2}\log(|A|) + c, \tag{12}$$

where $c$ represent the accumulated constant terms and

$$A = -\nabla\nabla(\log p(\boldsymbol{y}|\boldsymbol{f}) + \log p(\boldsymbol{f}|X, \boldsymbol{\theta}))_{|\boldsymbol{f}=\hat{\boldsymbol{f}}}. \tag{13}$$

The form of $p(\hat{\boldsymbol{f}}|X, \boldsymbol{\theta})$ is given in Equation 3 (in the main paper) and using the shorthand notation

$$\begin{aligned}
\boldsymbol{a} &= K_{XZ}K_{ZZ}^{-1}\boldsymbol{r}_{\boldsymbol{\theta}_r} \\
K &= K_{XX} - K_{XZ}K_{ZZ}^{-1}K_{ZX},
\end{aligned}$$

the log-likelihood in Equation 12 can be expressed as

$$\begin{aligned}
\ell(\boldsymbol{\theta}; \mathcal{D}) &\approx \log p(\boldsymbol{y}|\hat{\boldsymbol{f}}) - \frac{1}{2}\log(|A|) - \frac{1}{2}\log(|K|) \\
&\quad - \frac{1}{2}(\hat{\boldsymbol{f}} - \boldsymbol{a})^T K^{-1}(\hat{\boldsymbol{f}} - \boldsymbol{a}) + c.
\end{aligned} \tag{14}$$

The term $p(\boldsymbol{y}|\boldsymbol{f})$ in Equation 13 factorizes, hence $W = \nabla\nabla\log p(\boldsymbol{y}|\boldsymbol{f})$ is diagonal (Rasmussen & Williams, 2006, Page 43) with

$$\begin{aligned}
W_{ii} &= \nabla\nabla\log p(y_i|f_i) = \nabla\nabla\Phi(y_i \cdot f_i) \\
&= -\frac{(f_i)^2}{\Phi(y_i \cdot f_i)^2} - \frac{y_i \cdot f_i \cdot (f_i)}{\Phi(y_i \cdot f_i)}.
\end{aligned}$$

From Equation 14, $\nabla\nabla\log p(\boldsymbol{f}|X, \boldsymbol{\theta}) = K^{-1}$, therefore $A = -W - K^{-1}$. We finally approximating the log-likelihood (omitting constant terms) as

$$\begin{aligned}
\ell(\boldsymbol{\theta}; \mathcal{D}) &\approx \log p(\boldsymbol{y}|\hat{\boldsymbol{f}}) - \frac{1}{2}\log(|-W - K^{-1}||K|) \\
&\quad - \frac{1}{2}(\hat{\boldsymbol{f}} - \boldsymbol{a})^T K^{-1}(\hat{\boldsymbol{f}} - \boldsymbol{a}).
\end{aligned}$$

Lastly, we find $\hat{\boldsymbol{f}}$ using Newton's approximation for $T$ steps. Starting with $\hat{\boldsymbol{f}}^0 = \boldsymbol{0}$, for $1 \leq t \leq T$, we have

$$\hat{\boldsymbol{f}}^{t+1} = (K^{-1} + W)^{-1}(W\hat{\boldsymbol{f}}^t + \nabla\log p(\boldsymbol{y}|\hat{\boldsymbol{f}}^t)),$$

where $\nabla\log p(y_i|f_i) = y_i \mathcal{N}(f_i)/\Phi(y_i f_i)$.

## C  ADDITIONAL EXPERIMENTS ON REGRESSION TASKS

Following the setup described in (Chen et al., 2020), we compared our approach to PCG-based exact GP (EGP) (Wang et al., 2019), sparse GP regression (SGPR) (Titsias, 2009), stochastic variational GP (SVGP) (Hensman et al., 2013), and sgGP (Chen et al., 2020) on several simulated and real regression benchmark datasets. Briefly, EGP leverages GPU parallelization and conjugate gradients to compute the exact covariance matrix on large datasets, SGPR uses a variational formulation for sparse approximations that jointly infers the inducing inputs and the kernel hyper-parameters by maximizing a lower bound of the true log marginal likelihood, SVGP variationally decomposes Gaussian processes to depend on a set of globally relevant inducing variables, and sgGP exploits stochastic gradient descent on the Gaussian process likelihood to learn the kernel hyper-parameters.

## D  IMDB-TEXT INDUCING POINTS

We report the training set reviews close to the inducing points for the IMBD-Text experiment presented in Section 4.4 for one of the ten runs. Recall that we use 128 inducing points. In this case, we obtained 60 unique inducing points corresponding to 32 positive and 28 negative training set reviews. The inducing points are all different to each other but there are collisions when choosing the closests points in the training set. Note that the inducing points are balanced among the two classes. For the sake of space, for each review we report the initial and last 25 words.

Table 3: Comparison of root mean square error (RMSE) of different GPs on the benchmark datasets. We report the mean and standard error of RMSE. The best results are in bold (lower is better). For query and borehole datasets, EGP is not able to fit due to memory constraints.

| DATASET | SIZE | D | IGN (OURS) | sGGP | EGP | SGPR | SVGP |
|---|---|---|---|---|---|---|---|
| LEVY | 10,000 | 4 | $\mathbf{0.18}_{\pm\mathbf{0.01}}$ | $0.27_{\pm0.00}$ | $0.31_{\pm0.00}$ | $0.56_{\pm0.01}$ | $0.58_{\pm0.01}$ |
| GRIEWANK | 10,000 | 6 | $\mathbf{0.05}_{\pm\mathbf{0.00}}$ | $0.07_{\pm0.00}$ | $0.19_{\pm0.07}$ | $0.13_{\pm0.00}$ | $0.09_{\pm0.01}$ |
| BOREHOLE | 1,000,000 | 8 | $\mathbf{0.00}_{\pm\mathbf{0.00}}$ | $0.17_{\pm0.00}$ | — | $0.18_{\pm0.00}$ | $0.17_{\pm0.00}$ |
| PROTEIN | 45,730 | 9 | $\mathbf{0.64}_{\pm\mathbf{0.01}}$ | $0.66_{\pm0.01}$ | $0.69_{\pm0.00}$ | $0.72_{\pm0.00}$ | $0.68_{\pm0.00}$ |
| PM2.5 | 41,757 | 15 | $0.31_{\pm0.01}$ | $\mathbf{0.29}_{\pm\mathbf{0.00}}$ | $\mathbf{0.29}_{\pm\mathbf{0.00}}$ | $0.64_{\pm0.01}$ | $0.54_{\pm0.01}$ |
| ENERGY | 19,735 | 27 | $\mathbf{0.74}_{\pm\mathbf{0.02}}$ | $0.79_{\pm0.00}$ | $0.80_{\pm0.07}$ | $0.84_{\pm0.01}$ | $0.80_{\pm0.01}$ |
| BIKE-HOUR | 17,379 | 15 | $\mathbf{0.01}_{\pm\mathbf{0.00}}$ | $0.22_{\pm0.00}$ | $0.23_{\pm0.00}$ | $0.28_{\pm0.00}$ | $0.25_{\pm0.01}$ |
| QUERY | 100,000 | 4 | $0.08_{\pm0.00}$ | $\mathbf{0.05}_{\pm\mathbf{0.00}}$ | — | $0.06_{\pm0.00}$ | $0.06_{\pm0.00}$ |

1. **positive** - there have been several films about zorro some even made in europe e g alain delon this role has also been played by outstanding actors ... good performance of hadley as zorro he was quick smart used well his whip and sword and his voice was the best for any zorro

2. **negative** - i love the frequently masters of horror series horror fans live in a constant lack of projects like this and the similar project with gave ... up has to have a payoff that exceeds build up not the other way around storytelling math 101 br br end of spoilers big oops

3. **positive** - i rented the film i don't think it got a theatrical release here out expecting the worse the previews made the film look awful i ... be proclaimed 'the worst film ever i recommend this film for anybody interested in the show a flawed but innovative and interesting piece of film

4. **positive** - one of the best of the fred astaire and rogers films great music by irving berlin solid support from randolph scott harriet nelson lucille ball ... jazzy and it's a great song br br fun all the way although i got tired of we joined the navy after the third time

5. **positive** - forbidden planet rates as landmark in science fiction carefully staying within hard aspects of the genre science not fantasy nerds will love it while still ... the edge destroying its creator just as it did thousands of centuries earlier to the krell br br maybe the krell had teenage daughters too

6. **negative** - the bad news is it's still really dreadful i gave it a 2 because occasionally some of this slapstick parody actually seems funny br br ... this one and get this dvd back to the video store on time you'll really hate yourself if you have to pay a late fee

7. **positive** - nice character development in a pretty cool milieu being a male i'm probably not qualified to totally understand it but they do a nice job ... but within this world it needed to happen good acting all around with something positive taking place in the lives of some pretty good people

8. **negative** - mercifully there's no video of this wannabe western that a stay afloat vehicle for big frank at a time when his career was floundering the ... you up late and having a bout of insomnia but if you can sit through it you've more than most of my movie buff friends

9. **negative** - bela lugosi is an evil who sends brides poisoned on their wedding day steals the body in his fake ambulance hearse and takes it home ... in a discount store 2 for £1 which i think is a pretty accurate anyone paying more for this would be out of their mind

10. **positive** - its no surprise that busey later developed a in his this film is also a poor decision but one i enjoyed fully the first 5 ... wet myself some of best work by far rent or buy it today my vote is a perfect 10 on the poo meter that is

11. **positive** - before sky i saw diane tender performance in this otherwise of a movie campers are invited to the camp of their youth and experience it ... comic acting turn by noted director sam raimi makes this a movie you can pull out again and again like looking up an old friend

12. **positive** - the line of course is from the lord's prayer thy will be done on earth as it is in heaven sweden especially its far north ... the ending is what you make of it i guess but it's not spoiling it to say daniel achieves what he set out to do

13. **positive** - my certainly is a fair looking woman this film is a lost gem a dead on satire mockumentary of the early 90's hip hop scene ... this regard i regard this movie like the 1000 islands of upstate new york it's a wonderful little secret you want to keep to yourself

14. **positive** - i rented this film from netflix for two reasons i was in the mood for what i thought would be a silly '50s sci fi ... also generally very good and the acting is much better than one might expect i was particularly impressed with reeves jeff corey and walter reed

15. **negative** - brilliant book with wonderful characterizations and insights into human nature particularly the nature of addiction which still resonate strongly today br br as for the ... normally excellent but inappropriately cast actors all in all a weak adaptation your three hours would be better spent reading or re reading the book

16. **negative** - it's boggles the mind how this movie was nominated for seven oscars and won one not because it's abysmal or because given the collective credentials ... director hungry to be recognized it could've been morphed to something better but what's left looks like a film nobody was really interested in making

17. **positive** - a comedy of funny proportions from the guys that brought you south park and most of the guys from this movie has utterly disgusting and ... turn the sport sour and its up ta coop ta fix it and along the way you will laugh alot that's all there is enjoy

18. **negative** - there was a bugs bunny cartoon titled baby buggy bunny that was exactly this plot baby faced robbed a bank and the money in the ... the bugs bunny dvd it's was much more original the first time 1954 plus you'll get a lot more classic bugs bunny cartoons to boot

19. **negative** - first off i really loved henry fool which puts me in a very small pool of movie goers parker posey is one of best actresses ... ride i'd be happy to spoil this movie for you but it's been done it's rotten the fool franchise is dead long live henry fool

20. **positive** - i imagine victorian literature slowly sinking into the of the increasingly distant past pulled down by the weight of its under skirts along comes television ... coarse have been made to modern tastes and without having felt preached to another bbc classic highly recommended this is how romantic literature should be

21. **negative** - i knew this movie wasn't going to be amazing but i thought i would give it a chance i am a fan of luke wilson ... the movie without people getting annoyed the movie had its moments but i'm glad i didn't spend 9 50 to see it in the theater

22. **negative** - i have read several good reviews that have defended and the various aspects of this film one thing i see over and over is annoyance ... of good and terrible acting i would recommend it for a cheap thrill but hardly a diamond in the rough that is micro budget horror

23. **negative** - this movie is like the thousand cat and mouse movies that preceded it the following may look like a spoiler but it really just describes ... exiting the theater from a hollywood movie and if you have ever felt that way too heed my warning stay miles away from this movie

24. **negative** - i wonder who how and more importantly why the decision to call richard attenborough to direct the most singular sensation to hit broadway in many ... well michael douglas was in it true i forgot i'm absolutely wrong and you are absolutely right nothing like a richard attenborough michael douglas musical

25. **positive** - this movie will go down down in history as one of the greats right along side of citizen kane casablanca and on the waterfront someone ... do yourself and your family a favor and buy it immediately i'm still holding out hope for a special edition dvd one of these days

26. **negative** - this is high grade cheese fare of b movie kung fu flicks bruce wannabe lee is played by bruce li i think of course let's ... flashback for a scene just shown 3 minutes ago they must've thought that only one with attention disorder could fully understand this film br br

27. **negative** - the only previous gordon film i had watched was the kiddie adventure the magic sword 1962 though i followed this soon after with empire of ... them then again this particular version is further sunk by the tacked on electronic score – which is wholly inappropriate and cheesy in the extreme

28. **positive** - hilarious evocative confusing brilliant film reminds me of or holy mountain lots of strange characters about and looking for what is it i laughed almost ... watch on screen or at his big slide show smart funny quirky and outrageously hot make more films write more books keep the nightmare alive

29. **positive** - the villian in this movie is one mean sob and he seems to enjoy what he is doing that is what i guess makes him ... guess you can make up your own mind about the true ending i'm left feeling that only one character should have survived at the end

30. **negative** - okay what the hell kind of trash have i been watching now the mountain has got to be one of the most incoherent and insane ... good heroine this is the type of european horror film that could have been legendary if only someone had bothered to write a structured screenplay

31. **positive** - i found this movie to be very good in all areas the acting was brilliant from all characters especially ms stone and character just gets ... audience which was misled by some faulty terrible reviews about the movie before it even started you won't regret it if you go see it

32. **negative** - a lot of horror fans seem to love scarecrows so i won't be very popular in saying that i found it to be rather boring ... involving killer scarecrows to my knowledge apart from dark night of the scarecrow which is much better i would recommend that over scarecrows any day

33. **negative** - david mamet's film debut has been hailed by many as a real thinking man's movie a movie that makes you question everybody and everything i ... unfulfilled and if you like me predicted ahead of time that margaret was going to be conned you will find this revelation just as unsatisfying

34. **positive** - while this was a better movie than 101 dalmations live action not animated version i think it still fell a little short of what disney ... as so many disney films are here's to hoping the third will be even better still because you know they probably want to make one

35. **positive** - i think i read this someplace joe johnston director of the film and also one of the guys who founded industrial light and magic for ... first homer jr did not like the idea but he warmed up to it after the movie poster paperback novel came out and took off

36. **positive** - stephen king movies are a funny thing with me i either really love them or i loathe them some of the productions such as desperation ... very watchable and enjoyable adaption br br for uk readers this production has most recently been shown on sci fi and sky thriller horror channels

37. **positive** - if you're researching ufo facts then this video is very important the of the video is the comments made by buzz he is without a ... in details should not detour your from viewing this video if nothing else it is interesting and i recommend you watch with an open mind

38. **positive** - rupert friend gives a performance as prince albert that lifts the young victoria to unexpected levels he is superb as we know queen victoria fell ... believe for a minute she was victoria no real sense of period it may no have been her fault but her prince deserved the crown

39. **positive** - seeing moonstruck after so many years is a reminder of how sweet and funny this film was when it first appeared who knew that cher ... used to be at its best entertainment with no social significance whatsoever if they'd only lost that's along the way it would have been perfect

40. **negative** - i first learned of the wendigo many years ago in one of alvin scary stories books according to that story the wendigo after calling your ... to count br br anyway avoid it patricia clarkson and erik per sullivan dewey on malcolm in the middle have done far better than this

41. **positive** - well maybe not immediately before the rodney king riots but even a few months before was timely enough my parents said that they saw it ... but either way grand canyon is a great movie it kevin kline as my favorite actor also starring mary mary louise parker and alfre woodard

42. **positive** - finally a movie where the audience is kept guessing until the end what will happen well we all kind of know that the lives of ... his drug and sex addictions and a father who finally discovers exactly what happened the day of the robbery this movie will get you thinking

43. **negative** - wow i don't even really remember that much about this movie except that it stunk br br the plot's basically a girl's parents neglect her ... you do see it don't expect much 1 out of 10 br br seriously if you want a pokemon movie rent pokemon the first movie

44. **positive** - the group of people are travelling to in an awful bus led by a drunk conductor and his dumb son who likes to drive with ... bigger than him in the end the movie takes one turn and the trip becomes nothing but a swan s song of a dying country

45. **negative** - yes this movie is a real thief it stole some shiny oscars from just because politicians wanted another war hero movie to boost the acceptance ... if we consider this title a reasonable piece of the u s wars are cool genre you surely have much better movies to choose from

46. **positive** - i was going through a list of oscar winners and was surprised to see that this film beat butch cassidy and the sundance kid for ... by hoffman to take this role otherwise he may have been typecast after the graduate anyway this considered an all time great for a reason

47. **positive** - inspirational tales about triumph of the human spirit are usually big turn offs for me the most surprising thing about men of honor is how ... doesn't disappoint he creates a darkly funny portrait director george jr set out to make an old style flick and comes up with a winner

48. **negative** - this movie was god awful from conception to execution the us needs to set up a star wars site in this remote country this is ... gymnast star in real life i would probably kick him in the face after a double with 2 1 2 twists in the layout position

49. **positive** - i wouldn't call we're back a story simply a kiddie version of jurassic park i found it more interesting than that like the former it ... kind i would actually say that john goodman doing voice here is sort of a precursor to his voice work in monsters inc worth seeing

50. **negative** - chinese ghost story iii is a totally superfluous sequel to two excellent fantasy films the film delivers the spell casting special effects that one can ... a little extra money out of a successful formula they won't be able to do it again the cash cow is now dead as a

51. **negative** - the direction had clearly stated that this film's idea and plot is totally original however as to those who have read comic we can clearly ... watching this thus making this movie getting what it shouldn't have it has became one of the best budget films in china for this year

52. **negative** - let me start by saying that i understand that invasion of the star creatures was meant to be a parody of the sci fi films ... the double feature dvd with invasion of the bee girls that movie is academy award winning stuff in comparison with invasion of the star creatures

53. **positive** - knowing when to end a movie is just as important as casting directing and acting and it's nice to see when a director script get ... a mansion br br this is a great independent production and one that wastes little time getting going and it won't waste your time either

54. **negative** - italian born has inherited from her deceased lover karl an ultra modern and isolated house in the middle of the woods it's winter and she ... around for it i think i'll give it the benefit of the doubt as it's definitely not what i was expecting from this indie film

55. **negative** - is a horror comedy that doesn't really have enough horror or comedy to qualify as one or the other it has one scene that is ... the movie that is weaker in general plot and spine because of production values that just shows you how uninteresting i found the look of

56. **positive** - there were a lot of 50's sci fi movies they were big draws for the drive in theaters a lot of them were crappy even ... worried about the invisible monster forbidden planet is a movie a sci fi fan can watch several times and find something new with each viewing

57. **negative** - curiously it is rene eyes and mouth not buddy the that emerge as the focal point of buddy a jim henson pictures production through francis ... thompson needs a good pick up shot she gives rene another extreme close up i wonder what the lipstick budget was on this picture from

58. **positive** - now this is what i'd call a good horror with occult supernatural undertones this nice low budget french movie caught my attention from the very ... very confusing towards the end but redeems itself by the time it's over br br i thought his was a very good movie 8 10

59. **negative** - and ethel buffs too will love her loud vocals as the wicked witch but this cartoon sequel to the wizard of oz is bereft of ... sure baby boomers will get a charge from it since it has been out of for so long as a curiosity item just fair from

60. **negative** - this is another one of those vs insects features a theme that was popular in the late 70's only you can't really call it horror ... after having seen ants lacking suspense action thrills shocks and creepiness the only thing you'll be left with after seeing ants is an annoying itch

