# OpenReview forum: "Inducing Gaussian Process Networks"
_ICLR.cc/2023/Conference — Submitted to ICLR 2023_

### Official Review · Reviewer_MpyF · 2022-10-21

**Confidence:** 4
**Correctness:** 3
**Technical Novelty And Significance:** 2
**Empirical Novelty And Significance:** 1
**Recommendation:** 5

**Clarity, Quality, Novelty And Reproducibility:**

The paper is clearly written, reproducible (the authors provide code for the experiments, though I did not try to run this), and the quality is generally good. As mentioned above, I think the novelty is limited.


### Questions for the authors
1. You explicitly represent the inducing outputs $\mathbf{r}$ as the output of a function on the inducing inputs. Generally, however, in the sparse GP framework, the inducing outputs are integrated out. What is the reason for keeping them here? Further to this, since the inducing outputs are scalars, why not simply optimise these scalars directly?
2. You mention that the DKL method by Wilson et al. (2016) uses inducing inputs in the input space. However, my understanding of the method is that the inducing points are placed in the learnt feature space since they replace the base covariance matrix with the KISS covariance matrix (Eq. (8) in Wilson et al. (2016)). Where do you see that the inducing inputs are placed in the input space?
3. What is a "straw man method"?
4. You mention that inducing points add explainability to the model. How so? I understand that you can look at the inducing inputs (or the nearest real inputs), but what will this tell you?

**Strength And Weaknesses:**

### Strengths
Deep kernel learning is an interesting field for the GP community, so the paper should be interesting to that part of the ICLR community. The paper also builds on recent advances in mini-batch optimisation for GPs, which is an exciting approach, and the empirical results are quite impressive.

### Weaknesses
The main weakness, in my view, is the novelty, which I consider limited. Deep kernel learning with learnable inducing points in the learnt feature space has been explored several times (for instance by Bradshaw et al. (2017), which the authors cite, and Ober et al. (2021)), and the fact that the authors highlight this as a novelty is somewhat misleading, I think. The main thing, in my view, that distinguishes this paper from others is that the authors use mini-batch optimisation of the inducing point log-marginal likelihood from Snelson & Ghahramani (2005), whereas other papers (at least the ones I could find) seem to use a variational sparse approximation in the learnt feature space. This is interesting since the convergence of SGD for GP inference was only recently established (Chen et al., 2020), but it is still an incremental change. This is also reflected in section 2, which is simply a summary of sparse GP regression and classification.

With that in mind, I think the experimental section is a little uninteresting. DKL has been shown to be a powerful approach many times, so the good performance of the proposed method is not that surprising. Instead, given the incremental nature, I would have been more interested in seeing a comparison between this method and other ways of using inducing inputs in feature space. For instance, Bradshaw et al. (2017) use a variational sparse approximation and optimise the resulting ELBO. Is this approximation better or faster than the one proposed here? Ober et al. (2021) show that DKL is prone to overfitting; is that also the case here? Similarly, the reason for computing the inducing outputs is not clear to me (see the questions in the next section), and it would have been interesting to see a comparison of ways to handle the outputs (e.g., representing them with a function as proposed vs optimising their values directly vs integrating them out).


### References
- Bradshaw et al. (2017): https://arxiv.org/abs/1707.02476
- Chen et al. (2020): https://arxiv.org/abs/2111.10461
- Ober et al. (2021): https://arxiv.org/abs/2102.12108

**Summary Of The Paper:**

The paper presents a deep kernel learning method for Gaussian processes. Deep kernel learning parametrises the covariance function using a neural network, and the proposed method further optimises a set of inducing points in the transformed space as well as employs stochastic gradient descent, allowing for faster inference. The authors demonstrate the expressiveness of their method through experiments on tabular data for regression, image classification, and classification of text and graphs.


**Summary Of The Review:**

The proposed method is a variation of deep kernel learning for Gaussian processes. The novelty seems to be that the authors use mini-batch stochastic gradient descent to optimise the inducing point log-marginal likelihood from Snelson & Ghahramani (2005) instead of a variational objective used in previous work. This is an incremental change, however, and the experimental section is a little off in my opinion; instead of only showing that the method performs well on a range of tasks, there should have been a focus on comparing it to the other strongly related methods, trying to understand the strengths and weaknesses of each approach.

Deep kernel learning as a topic is interesting to the GP community broadly, and the proposed method does achieve good empirical results, which should make it useful for practitioners as well as a strong baseline for future papers. However, given its current focus (and, in my opinion, incorrectly claimed novelty), I think it's not strong enough for acceptance.

---

### Official Review · Reviewer_r62e · 2022-10-24

**Confidence:** 3
**Correctness:** 4
**Technical Novelty And Significance:** 3
**Empirical Novelty And Significance:** 2
**Recommendation:** 5

**Clarity, Quality, Novelty And Reproducibility:**

- Although I found the problem statement to be both interesting and relevant, the contributions currently rely too heavily on obtaining empirical gains, whereas the overall theoretical motivation is less complete.
- The paper is well-written overall and fairly easy to follow. There are some aspects of the derivation which I think could be clarified further however, such as the role and selection of the pseudo-label function.
- There are a few typos and minor careless mistakes in the paper’s writing, but these should be fairly easy to iron out in a future revision of the paper. The references need to be properly cleaned however - certain words (e.g. Gaussian) need to be consistently capitalized across paper titles, and the detail included for citations also needs to be properly homogenized.
- There appears to be sufficient detail included in the main paper and supplement for reproducing the featured experiments. Appropriate tables and plots with error bars were also included throughout.

**Strength And Weaknesses:**

- The paper identifies an interesting limitation of existing inducing point approaches, and sets up a clear problem statement that is well framed in the context of related work. Showing how the proposed model can be extended to classification tasks via the Laplace approximation also makes the paper come across as more complete.
- I appreciated how the paper considered an extensive selection of varied datasets in the experimental evaluation, which effectively demonstrates how the proposed IGNs can be used by practitioners across different use cases. The ablation studies assessing how the variance changes as more inducing points are added is also interesting, if not entirely surprising.

------

- I believe that some of the experiments would benefit from a more thorough deep dive into why certain results were obtained. For example, IGNs appear to perform quite inconsistently on the tabular regression datasets where the method varies from over-performing on some datasets while comparing less favorably to competing techniques on others.
- I expect there to be a substantial risk of overfitting using this scheme, especially compared to methods using variational inference to optimize the location of inducing points. I believe this to be a particularly important consideration for practitioners, however I didn’t find much discussion in the paper.
- I would have liked to see the method be framed in a similar context to the discussion featured in _“Understanding probabilistic sparse gaussian process approximations”_ by Bauer et al (2016) which presents a more principled discussion of how different inducing point GP approximations compare against each other. Such insight is currently missing from this work.
- Although the paper stresses improved computational complexity as being one the primary benefits of using IGN models, this comparison is currently relegated to the supplementary material, whereas I believe this would also have been interesting to feature in the main paper.


**Summary Of The Paper:**

Inducing points have traditionally been heavily relied on to alleviate the cubic computational complexity associated with Gaussian process training and inference. There have been various different ways in which inducing points have been used in the literature, but they have nearly always been limited to either the input space of the data, or else were constrained by either the positions (e.g., to form a grid structure) or choice of embedding (such as trigonometric representations of the input space). In this work, the authors propose an approach for learning the inducing points along with the deep kernel function itself, which no longer has to be pre-trained. This allows for greater flexibility in the adaptation of inducing points since they are not constrained by the complexity or structure of the input space. The authors demonstrate how the proposed Gaussian process network models (IGN) achieve comparable performance to existing techniques on standard tabular and image data, while also being easily extendable to datasets having more specialized structures, such as text and graphs.

**Summary Of The Review:**

I think there are some solid ideas in this paper that are backed by a fairly extensive experimental evaluation. Nevertheless, I feel quite ambivalent about the paper’s novelty and contributions overall, and would like to see more depth and insight included in a future revision of the paper. I am presently borderline on this work, tending towards rejection in the belief that there is still considerable room for improvement on the current submission.

---

### Official Review · Reviewer_dTCT · 2022-10-25

**Confidence:** 3
**Correctness:** 3
**Technical Novelty And Significance:** 2
**Empirical Novelty And Significance:** 3
**Recommendation:** 5

**Clarity, Quality, Novelty And Reproducibility:**

- This manuscript is well-written.
- Technical novelity is somewhat low.
- Although the authors have experimented with a variety of datasets, some concerns remain.

**Strength And Weaknesses:**

S1. The authors proposes a scalable learning method for Gaussian processes with deep kernel. The proposed method is simple but useful.

S2. Experiments demonstrate the effectiveness of the proposed method on various datasets.

W1. Although the proposed method is practically useful, it seems a straightforward combination of inducing points and deep kernel learning.

W2. There is no discussion of predictive variance. It would be good to find out how accurately it can be estimated by the proposed method.

W3. There is still some concern about the number of inducing points, although an ablation study on it has been done in Section 5. This hyperparameter may be quite difficult to determine, since the appropriate value may vary depending on the difficulty of the problem and the dimension of z.

**Summary Of The Paper:**

This paper proposes a new Gaussian process model that combines inducing point technique and deep kernel learning. The proposed method can be used for regression and classification. Experimental results verify the effectiveness of the proposed method on various types of real data.

**Summary Of The Review:**

I agree that the proposed method is simple but useful. But the technical novelity seems moderate.

---

### Decision · Program_Chairs · 2023-01-20

**Decision:**

Reject

**Justification For Why Not Higher Score:**

The main weakness is the novelty of the paper. Deep kernel learning with learnable inducing points in the learnt feature space has been explored several times, and the fact that the authors highlight this as a novelty is somewhat misleading,


**Justification For Why Not Lower Score:**

N/A

**Metareview: Summary, Strengths And Weaknesses:**

The paper presents a deep kernel learning method for Gaussian processes. Deep kernel learning parametrises the covariance function using a neural network, and the proposed method further optimises a set of inducing points in the transformed space as well as employs stochastic gradient descent, allowing for faster inference. The authors demonstrate the expressiveness of their method through experiments on tabular data for regression, image classification, and classification of text and graphs.

The main strength is the paper identifies a critical limitation of existing inducing point approaches and uses the mini-batch optimization of the inducing point likelihood.

The main weakness is the novelty of the paper. Deep kernel learning with learnable inducing points in the learnt feature space has been explored several times, and the fact that the authors highlight this as a novelty is somewhat misleading,